# Bayesian nonparametric models characterize instantaneous strategies in a competitive dynamic game

Kelsey R. McDonald [1,2,3], William F. Broderick [4], Scott A. Huettel [1,2,3] & John M. Pearson [1,2,3,5]

Previous studies of strategic social interaction in game theory have predominantly used games with clearly-defined turns and limited choices. Yet, most real-world social behaviors involve dynamic, coevolving decisions by interacting agents, which poses challenges for creating tractable models of behavior. Here, using a game in which humans competed against both real and artificial opponents, we show that it is possible to quantify the instantaneous dynamic coupling between agents. Adopting a reinforcement learning approach, we use Gaussian Processes to model the policy and value functions of participants as a function of both game state and opponent identity. We found that higher-scoring participants timed their final change in direction to moments when the opponent's counter-strategy was weaker, while lower-scoring participants less precisely timed their final moves. This approach offers a natural set of metrics for facilitating analysis at multiple timescales and suggests new classes of experimental paradigms for assessing behavior.

[1] Duke Institute for Brain Sciences, Duke University, Durham 27710 NC, USA. [2] Center for Cognitive Neuroscience, Duke University, Durham 27710 NC, USA. [3] Department of Psychology and Neuroscience, Duke University, Durham 27708 NC, USA. [4] Center for Neural Science, New York University, New York 10003 NY, USA. [5] Department of Biostatistics and Bioinformatics, Duke University Medical School, Durham 27710 NC, USA. Correspondence and requests for materials should be addressed to J.M.P. (email: john.pearson@duke.edu)

Over the last 15 years, game theory has been foundational in establishing cognitive and biological mechanisms of strategic decision making[1–4]. Paradigms like Matching Pennies, Trust/Ultimatum Games, and Prisoner's Dilemma have used simple choices in highly standardized contexts to yield key insights into social decision-making in humans and animals[3–12]. These game theory paradigms draw upon a vast literature detailing how rational players would behave[1,4,5,13–15], yet studies comparing human behavior to these normative solutions have found that humans often violate rational predictions[1,4,13,16].

While a central aim of game theory is to describe how people should make decisions, describing how humans actually make decisions is of particular interest to social scientists. Indeed, many of the features that have made game theory paradigms analytically attractive—discrete choices, turn-taking, known payouts—are abstractions away from real-world social interactions. For instance, when buyers haggle over the price of a good, they interact in real time, using a combination of nonverbal cues, strategic planning, perspective taking, and value judgment. Their continuous, dynamic interaction thus poses a challenge to any computational framework for the study of social decisions[4,17,18]. Moreover, while game theory has made progress in generalizing optimal strategies for games in continuous time and space, such as duels and auction bidding[4,14,19–22], considerably less has been done to quantify highly dynamic behavior in cases where optimal strategies remain challenging to compute. While game theory has proven highly successful in analyzing various sorts of equilibria players might settle into, considerably less is known about the processes by which these equilibria are reached[23,24]. As a result, it is desirable to develop analytical tools capable of quantifying strategic dynamics while maintaining the mathematical rigor that has made game theory such a productive framework.

Here, we introduce a computational modeling framework that borrows from recent advances in reinforcement learning[25–31], game theory[14,20,21], and nonparametric Bayesian modeling[32–34] to capture these social dynamics. Our approach produces models of behavior that are both flexible enough to capture the variability present in a continuously evolving strategic setting and powerful enough to quantify strategic differences across participants, trials, and even individual moments within trials. Our testbed for these ideas is a competitive task in which human participants played against both a human opponent and a computer opponent in a real time, movement-based game. This paradigm generates a rich complexity in individuals' behavior that can be succinctly described by individualized, instantaneous policy and value functions, facilitating analysis at multiple timescales of interest. This approach quantifies complex interactions between multiple agents in a parsimonious manner and suggests new classes of tractable paradigms for studying human behavior and decision making.

## Results

### Penalty shot task
We adapted a zero-sum dynamic control task, inspired by a penalty shot in hockey[18]. The task was viewed on a computer screen and played by two players: an experimental participant ($N = 82$) who controlled an on-screen circle, or puck, and another long-term participant who controlled an on-screen bar, acting as the goalie. Hereafter, we will refer to these players as the participant and the opponent, respectively. The puck began each trial at the left of the screen and moved rightward at a constant horizontal speed. The participant's objective was to score by crossing a goal line located at the right end of the screen behind the opponent. The opponent's task was to block the puck from reaching the goal line. Each player moved his or her avatar using a joystick. Both players were only able to control the vertical

velocities of their respective avatars, though the puck and bar had distinct game physics (see Methods and Supplementary Methods). See Fig. 1a, b for task progression and sample trajectories, as well as the Supplementary Movie 1 for a movie demonstrating real game play.

Participants played the penalty shot task in an fMRI scanner. Here, we report only the behavioral data from this experiment. On each trial, participants played against a randomly selected opponent. On half of the trials, this was a human opponent, located outside the scanner (each participant interacted with only one human, but two long-term human participants played as the goalie through data collection). On the other half of trials, the opponent was a computer algorithm. This opponent followed a track-then-guess heuristic in which it attempted to match the puck's vertical position (with a variable reaction time) before randomly choosing a direction to move at maximal speed near the end of the trial. This choice was motivated not only by pilot data that showed such a strategy was difficult for participants to exploit, but also by past work analyzing the anticipatory strategies of goalkeepers[35–37]. When subjects were asked after the experiment which opponent had a better strategy, half of the subjects ($N = 41$) reported that they thought the human opponent was better, and the other half reported that the computer algorithm was better, suggesting that the computer algorithm did, in fact, play at a level comparable to both human opponents.

As expected, participants exhibited considerable variability in game play. Figure 1 shows all trajectories for a representative pair of subjects. A salient feature of our paradigm is its accommodation of widely varying individual strategies. For each of our main analyses, we only display findings for a subset of participants; plots for all representative subjects for all analyses are available in Supplementary Figs. 4–11. Trajectories varied widely both within and across participants, despite the fact that players each only had one continuous degree of freedom (position along the y-axis). For example, Participant 3 (Fig. 1c) demonstrated highly stereotyped play, with most trials exhibiting a "down-up-guess" approach. By contrast, Participant 4's (Fig. 1d) trajectories were dispersed throughout the screen, perhaps resulting in less predictable play. Participants also experienced highly variable win rates, which ranged from 43–76% (against human: 34–83%; computer 42–73%).

### Gaussian process models
Observed data for each trial were movement trajectories for the puck and the bar, each spanning approximately 1.5 s (94–96 discrete time points). While it is possible to model these time series directly[18], we observed that puck trajectories could be reduced to a series of straight-line segments of near-maximal velocity separated by change points (Fig. 2a). That is, we could redefine the decision available to the participant at each moment as whether or not to switch direction. This transforms a time-series modeling problem into a more tractable change point prediction problem, for which our predictors are a small number of game state variables.

Viewed through the lens of reinforcement learning, the decision of whether to switch direction at time $t$ is an action, $a_t$, and the probability of this action given a state of the world $\mathbf{s}_t$ is given by the policy function: $\Pi(a_t, \mathbf{s}_t, \omega) = p(a_t|\mathbf{s}_t, \omega)$, where we let $\mathbf{s}_t$ denote a vector of predictors at each timepoint and $\omega$ is a binary variable indicating the opponent's identity (computer = 0, human = 1)[25]. In our case, we define the action space as a single binary variable, with 1 indicating a change in direction and a 0 indicating continuation along the current trajectory. However, the state $s$ remains continuous and includes 7 predictor variables: the $x$ and $y$ positions of the puck, the $y$ position of the bar, their

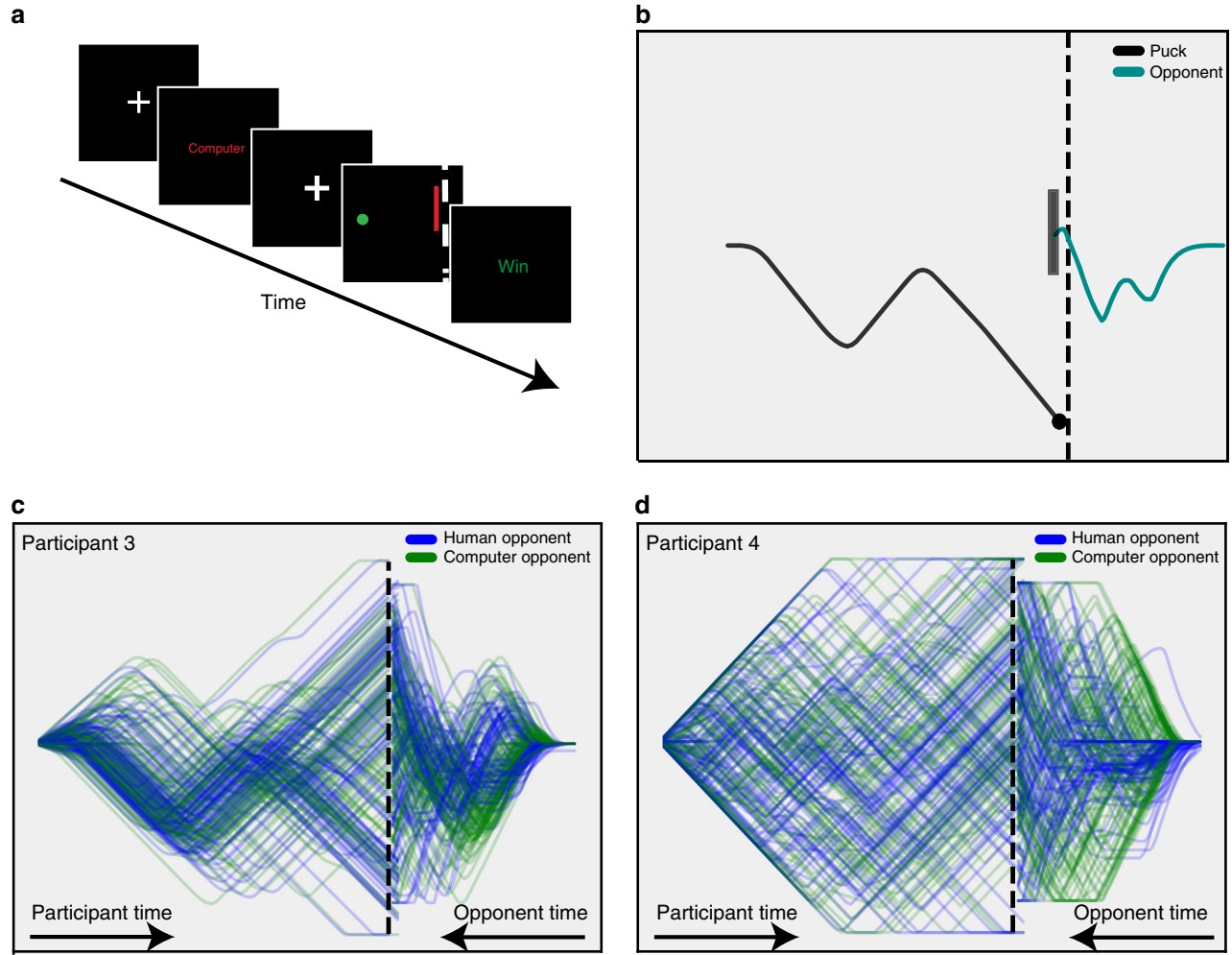

**Fig. 1** Strategic heterogeneity in dynamic decision-making. **a** Task progression: Following a jittered fixation cue, text indicated the identity of the opponent on the upcoming trial for 2 s. Play commenced after a variable delay during which the screen displayed a fixation cue. At the conclusion of each trial, which lasted roughly 1.5 s, colored text indicated the winner (green "Win" if the participant won; red "Loss" if the participant lost) for 1.5 s. **b** Game play on a single trial. The puck moves from left to right at constant horizontal velocity. The bar was only allowed to move vertically, but is depicted as moving from the right side of the screen inward toward the goal line for visualization purposes. **c**, **d** All of the trajectories for Participant 3 (**c**) and Participant 4 (**d**), demonstrating the heterogeneity observed across participants. Note variability in both on-screen positions' visited and trajectory shape: Participant 3 is much more consistent in game play, while Participant 4 was more variable. Trials played against the human opponent are displayed in blue. Trials played against the computer opponent are in green

respective vertical velocities, the time since the occurrence of the last change point (normalized to 1 by dividing by total trial length), and an opponent experience variable that ranged from 0 (first trial) to 1 (last trial) that was specific to each opponent and reflected potential strategic adaptation over the course of the experiment. Finally, we simplify our notation, defining $\pi(\mathbf{s}_t, \omega) = p(a_t = 1 | \mathbf{s}_t, \omega)$. Because our input space is of moderate dimension, a model for $\pi(\mathbf{s}, \omega)$ will be a continuous function of $\mathbf{s}$ instead of a large matrix, as it would be for a model with a discrete state space. Our contribution is to show that nonparametric methods allow us to address the challenge of modeling $\pi$ using only sparsely sampled data.

Our decision to model change point probabilities as a function only of states and opponents means that the data at each time are independent of each other given these variables. Thus, our approach is also equivalent to a binary classification problem. Binary classification is well-studied, with many methods available, including logistic regression, support vector machines, and neural networks[38]. Our model selection was guided by three requirements: First, the model should be flexible enough to capture the

rich diversity of player behavior. Second, the model should appropriately handle a small number of change points ($\approx$4.6%) with an input space of moderate dimension. Third, the model should avoid overfitting while providing a principled estimate of uncertainty. For these reasons, we fit each participant's data using a Gaussian Process (GP) classification model.

A Gaussian Process (GP) is a distribution over functions. GPs are widely used in spatial and time series modeling for their combination of flexibility and ability to generalize from even modest data[32,39]. In the same way that a sample from a normal distribution is a real number and a sample from a Bernoulli distribution is a binary variable, a sample from a GP is an entire function (e.g., a univariate time series ($d = 1$) or spatial density ($d = 2$)). Gaussian Processes have the advantage of providing a principled, Bayesian measure of uncertainty over functions[32], and while some types of GPs are equivalent to infinitely-wide, single-layer neural networks, they have been shown to outperform neural networks in avoiding overfitting on small to moderate datasets[32,40–42]. Moreover, they are the method of choice when modeling time courses based on sparse or irregularly-sampled

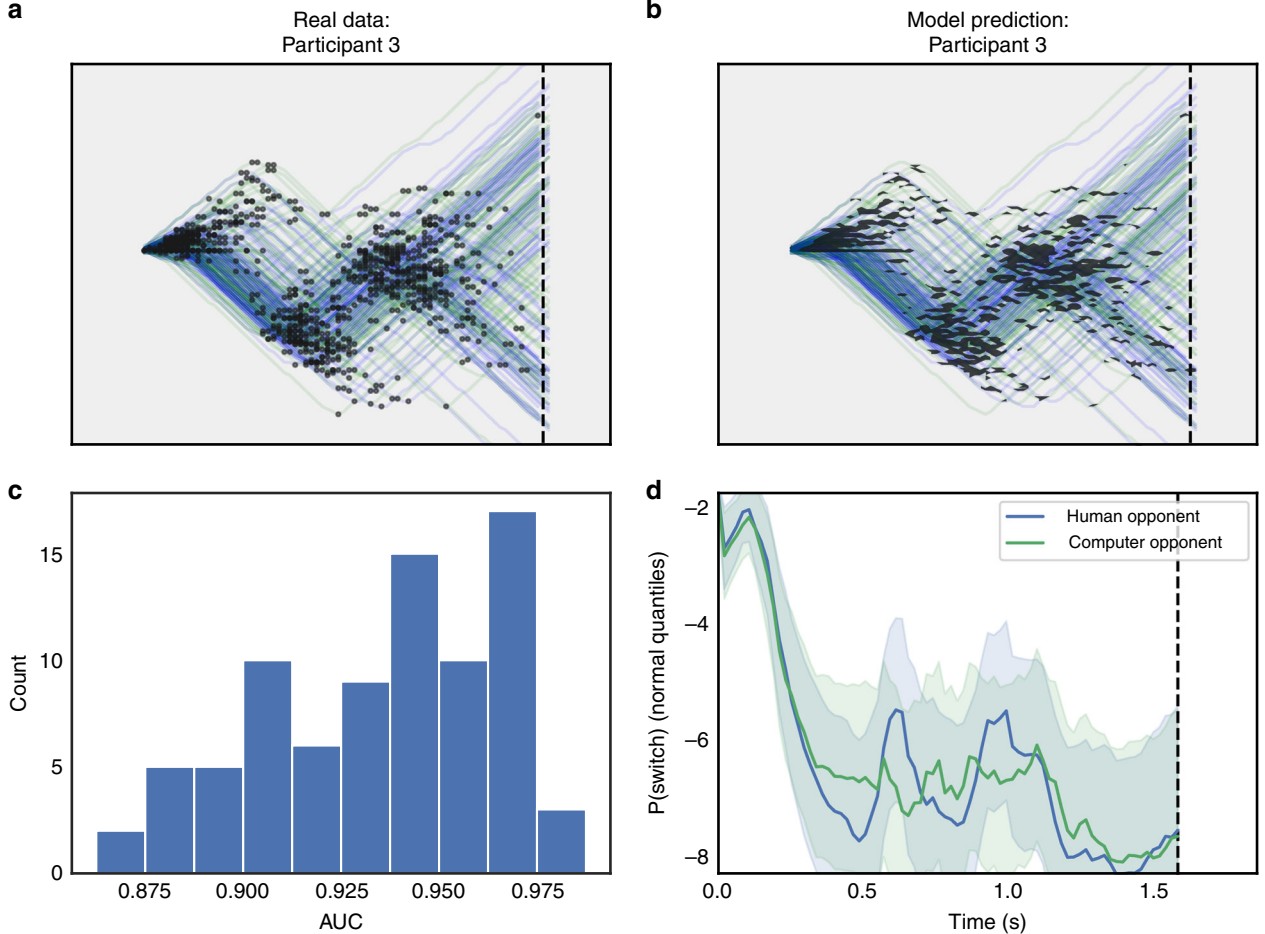

**Fig. 2** Change point classification captures strategy variability. **a** Observed data from a single subject (Participant 3) in the penalty shot task. Blue trajectories correspond to trials played against the human opponent. Green trajectories are from trials played against the computer. Black dots represent change points, or switches in joystick direction by the participant. **b** Same trajectories from A, but overlaid with black transparently-shaded regions indicating locations in which the model-predicted probability of a change points exceeded the participant's base rate. **c** Histogram of participants' area under the curve (AUC) scores on held-out data (20% of each participant's dataset). **d** Probability of a change point as a function of time, averaged across trials, for Participant 3. Shaded regions indicate 95% Bayesian credible intervals. Probabilities are shown and averaged in quantiles (z) of the normal distribution. Blue indicates trials against the human opponent, green against the computer

data[43,44]. Thus, GPs offer competitive modeling performance with the added benefits of uncertainty estimation and differentiability.

More formally, a GP is fully characterized by a mean function $m(\mathbf{x})$ (usually assumed to be 0 a priori) and a covariance function $k(\mathbf{x}, \mathbf{x}')$ that defines the correlation between values of $f$ at different input points[32]:

$$f(\mathbf{x}) \sim \mathcal{GP}(m(\mathbf{x}), k(\mathbf{x}, \mathbf{x}')) \tag{1}$$

$$m(\mathbf{x}) = \mathbb{E}[f(\mathbf{x})] \tag{2}$$

$$k(\mathbf{x}, \mathbf{x}') = \text{cov}[f(\mathbf{x})f(\mathbf{x}')] \tag{3}$$

where $f(\mathbf{x})$ is a random function drawn from the GP. By definition, the joint distribution of the observed dataset $\mathcal{D} = \{f(\mathbf{x}_i) | i = 1 \dots d\}$ is multivariate normal with dimension $d$, mean $\mu_i = m(\mathbf{x}_i)$, and covariance $\Sigma_{ij} = k(\mathbf{x}_i, \mathbf{x}_j)$.

As stated above, we chose to model players' policies via a GP classification model that predicted an upcoming change in the puck's direction from the current state $s$ and opponent identity $\omega$. Following standard techniques[32,45], we assumed that binary change point observations $a_i$ were Bernoulli distributed according to the policy $\pi(s, \omega)$ and that the policy itself was related to an underlying GP:

$$a \sim \text{Bernoulli}(\pi(\mathbf{s}, \omega)) \tag{4}$$

$$\Phi^{-1}(\pi) \equiv f(\mathbf{s}, \omega) \sim \mathcal{GP}(0, k) \tag{5}$$

where $\Phi^{-1}$ is the inverse cumulative normal distribution (also called the probit or quantile function) and $\mathcal{GP}(0, k)$ is a GP prior on $f$ with mean 0 and kernel function $k$. Because we assume that $f$ is a smooth function of its inputs, we choose the common radial basis function (RBF) kernel[32]:

$$k(\mathbf{x}, \mathbf{x}') = \sigma^2 \exp\left(\sum_{i=1} \frac{(x_i - x_i')^2}{\lambda_i^2}\right) \tag{6}$$

with $i$ indexing input variables and $\sigma_i$ and $\lambda_i$ hyperparameters setting the overall magnitude of the covariance and the length scale of correlations along each dimension, respectively. Here, $\mathbf{x}$ includes both $\mathbf{s}$ and $\omega$. Even though $\omega$ is a discrete parameter, we approximate it as a continuous variable, as is often done in Bayesian modeling using GPs[46].

We found that our GP classification model accurately captured the diverse patterns present in participants' data (Fig. 2a, b, d). That is, the model had a higher probability of predicting a change point in regions of the screen where change points actually

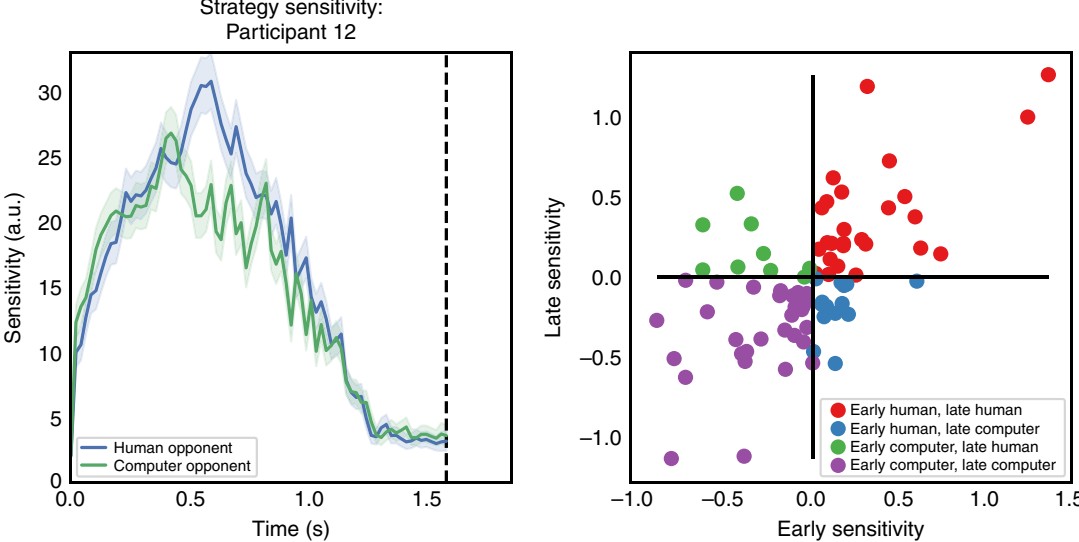

**Fig. 3** Sensitivity metric reveals information about player strategies. **a** Observed sensitivity to opponent actions in both opponent conditions, for Participant 12. Shaded regions indicate 95% credible intervals. Blue line and shaded region correspond to the human opponent condition, green to the computer opponent. **b** Differences in opponent sensitivity across participants. Scatter plot of relative opponent sensitivity for each participant in the dataset for early (horizontal) and late (vertical) phases of the trial. Values are differences in log sensitivity to each opponent, averaged across timepoints and trials. Positive values indicate higher sensitivity (either early or late) to the human opponent; negative values indicate higher sensitivity to the computer. Most participants (71%) demonstrate consistently higher sensitivity to one opponent throughout the trial (denoted by red and purple points)

occurred. This is a direct result both of the nonparametric nature of the GP—the model adapts its complexity to the data—as well as the smoothing effects of the prior. Held-out test data from each participant yielded a median area under the curve (AUC) score of 94% (Fig. 2c). For comparison, we also fit a regularized logistic regression for each subject, but for no subject did it outperform our GP model (see Supplementary Fig. 12).

Model fits revealed that participant policies were most strongly affected by the participant's own velocity, followed by the time since the previous change point and the participant's own vertical position (median length scales: 0.18, 0.23, and 0.43, respectively; standard deviations: 0.08, 0.40, and 0.71, respectively). That is, participants were more weakly influenced by their opponent's position and velocity than by their own movements, suggesting that their strategies were only secondarily reactive. Moreover, hyperparameters for the opponent experience variable, which captured changes in strategy over the course of the experiment, were large, indicating that their strategies quickly stabilized and remained consistent throughout play. In fact, trajectories for most players did not differ markedly in shape between the first and last ten trials (Supplementary Figs. 18 and 19).

**Sensitivity to opponent actions**. We next sought to quantify how much participants' switching behavior changed as a function of the opponent's actions. Because our change point policy model is based on a smooth Gaussian Process, we can quantify this sensitivity using gradients of the GP $f = \Phi^{-1}(\pi)$ with respect to the opponent's position and velocity (see Methods). We then used these gradients to define a moment-by-moment sensitivity index. Since the gradients of the GP measure the degree to which small changes in the current game state affect the participant's probability of changing course, gradients with respect to the opponent's position and velocity capture the degree to which the participant's current behavior is sensitive to the opponent's actions.

Just as switch probability changes dynamically with game state, sensitivity to opponent action varies throughout the trial. Figure 3a illustrates this for a single subject. In order to quantify

this, we asked whether participants' sensitivity differed depending on opponent, and whether this effect changed during the trial. As shown in Fig. 3b, most subjects (71%) were consistently more sensitive to one of the opponents in both phases of the trial. The few subjects who appear in the off-diagonal quadrants exhibited greater sensitivity to one opponent early and the other late. Thus, sensitivity-based metrics not only offer a precise characterization of variability in player strategies, but they also capture differences in play against each opponent.

**Sensitivity metrics characterize behavior**. The sensitivity metric defined above represents a particular moment-by-moment measure of the degree to which one player (the participant) is coupled to the actions of the other (the opponent). Based on our prior expectation, we chose a combination of sensitivities to opponent position and velocity, but other combinations are equally plausible. In fact, one could define a sensitivity metric to each input variable individually. Here, we show that such an approach produces a principled characterization of participants' behavior across multiple timescales. Indeed, when aggregated at the participant level, these indices fully characterize the policy model.

We defined one sensitivity for each input variable equal to the square of the gradient along each input direction (see Methods). This yielded eight new sensitivity indices (seven for state plus one for opponent identity) in addition to the opponent action sensitivity defined above. However, our previous index can be defined in terms of these new indices, so there are only eight unique values in the set. The most important feature of these new indices is that, like the policy, they are defined moment-by-moment, but can be aggregated across multiple levels of granularity, including trial and participant averages. As in classic analysis of variance (ANOVA), we can consider each index value at each data point as the sum of three terms: a participant-level mean, a trial-level offset from this mean, and a residual specific to the data point. Likewise, we can estimate variances within trial (residual), across trials, and across our participant population. As in ANOVA, the sum of these variances, appropriately weighted, equals the total variance in the data. Normalizing by this total

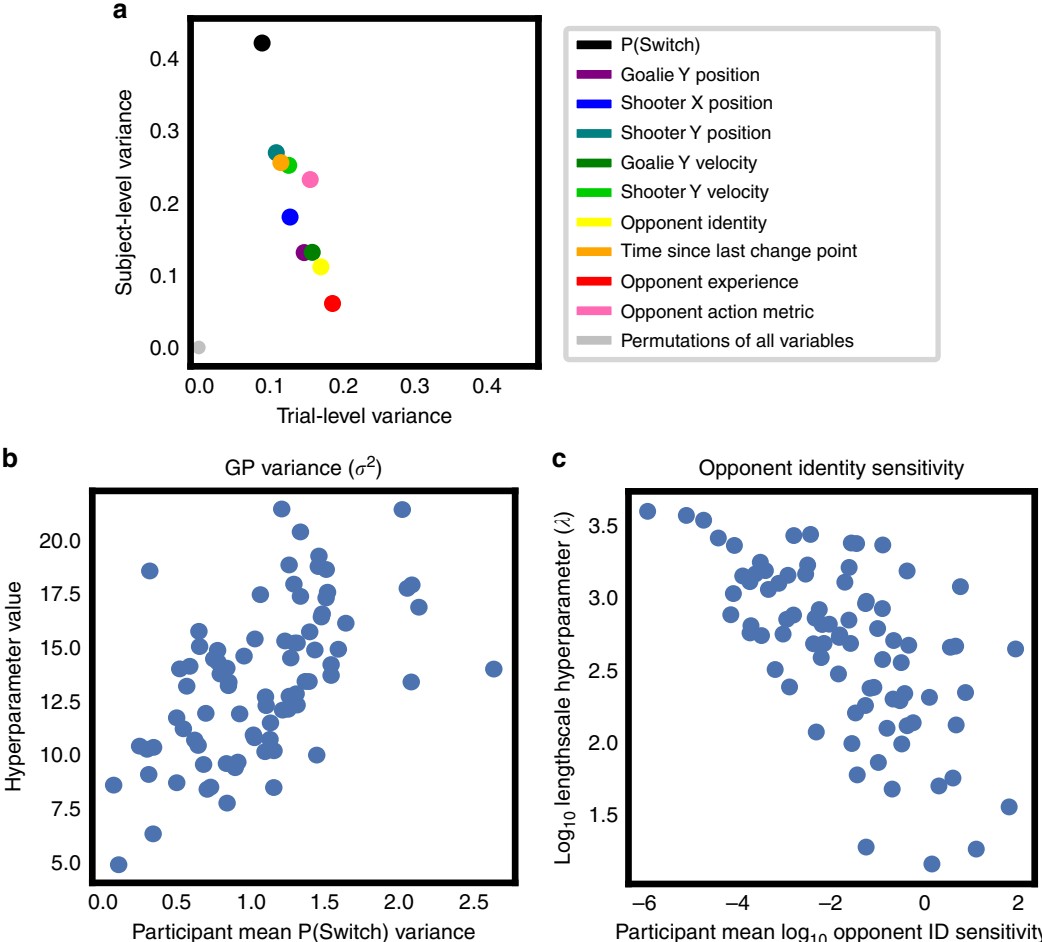

**Fig. 4** Sensitivity metrics describe behavior across many timescales. **a** Variance decomposition of model metrics. Proportions of trial- and subject-level variance for each predictor variable. Variables high in across-participant variance represent more trait-like variables (upper left), while variables high in variance across trials (lower right) are more state-like. When trial and subject label information are randomly permuted, the resulting indices account for ≈0% variance at the trial and subject levels (gray dots). **b** Relationship between mean probability of switching and GP noise parameter across participants. Each dot represents one participant. **c** Relationship between sensitivity to opponent identity and GP opponent identity hyperparameter

variance yields a set of three positive terms that sums to 1:

$$\frac{\sigma^2_{\text{participants}}}{\sigma^2_{\text{total}}} + \frac{\sigma^2_{\text{trials}}}{\sigma^2_{\text{total}}} + \frac{\sigma^2_{\text{residual}}}{\sigma^2_{\text{total}}} = 1 \qquad (7)$$

To illustrate this decomposition, we plot the trial- and subject-level variance for each predictor variable in Fig. 4a. We find that the subjects' strategies, as summarized by the baseline probability of switching, exhibit the largest variance across participants, indicating that the differences in change point frequency apparent in Fig. 1c, d are relatively more trait-like than our sensitivity measures. Perhaps surprisingly, while the sensitivities to opponent position and velocity demonstrate relatively little across-subject variance, the aggregated metric defining opponent action sensitivity has more across-subject variability, suggesting a more stable index. To test whether these effects could be the result of chance, we ran a permutation test in which we performed the variance decomposition analysis 1000 times with the trial and subject labels shuffled. These results, plotted in gray, tightly cluster around 0% for every variable. That is, our sensitivity indices in all cases account for significant variance in the data.

Finally, we note an important relationship between participant-level sensitivities and model hyperparameters. Because Gaussian Processes, like Gaussian distributions, are completely

characterized by their mean and covariance, summaries of a Gaussian Process taken over the data can only be functions of the hyperparameters that define the mean and covariance. That is, we expect on mathematical grounds that our sensitivities, when averaged across an entire participant's data, should be related to the model's hyperparameters (see Supplementary Note 2). Figure 4b, c illustrates this relationship for two hyperparameter-sensitivity pairs. Figure 4b shows that the noise parameter of our Gaussian Process model, $\sigma^2$ is indeed correlated with the variance in probability of switching across timepoints for each participant ($R = 0.56$, $t = 6.09$, $p < 0.0001$). Likewise, Fig. 4c shows that the logged hyperparameter controlling opponent identity effects in the GP correlates negatively with each participant's sensitivity to the same variable ($R = -0.64$, $t = 7.43$, $p < 0.0001$). In both cases, this is exactly what we expect: the noise hyperparameter for a classification model is related to the variance in its predictions, while low sensitivities correspond to long correlation length scales. Thus, our gradient-based sensitivity metrics naturally extend GP hyperparameters to the timepoint level, providing a principled characterization of strategy suitable for analysis at multiple timescales.

**Expected value of making one's final move**. We have shown that we can use nonparametric methods to estimate the policy

participants use when playing a dynamic, strategic game. Yet this analysis says nothing about how effective these policies are. So how do participants' choices at each moment translate to wins and losses? To answer this, we separately modeled each participant's action value $Q_\pi(a|\mathbf{s}, \omega)$: the expected value of taking action $a$ in state $s$ against opponent $\omega$ and playing according to policy $\pi$ thereafter. As indicated by notation, this value is policy-dependent. That is, each policy $\pi$ uniquely determines a value function $Q_\pi$ (see Supplementary Fig. 2a). In typical reinforcement learning models, policies are likewise dependent on action values: Given action values, $Q$, policies choose actions based on a softmax function or other rule[25]. Thus, there is a mapping in the reverse direction from action values to policies. The Bellman Equation stipulates that for optimal learners, the optimal policy and action values determine one another[25], but this need not hold for nonoptimal learners.

To capture this distinction, we also modeled each subject's empirical action value function $Q(a|\mathbf{s}, \omega)$. The action value model took as inputs the instantaneous state, opponent, and observed action at that time and attempted to predict from those data whether the participant subsequently won the trial. We used the same Gaussian Process classification approach as before, only this time predicting the trial outcome and using the participant's observed action (i.e., did the participant make a direction change at the next timepoint) as an additional input. Results from this model are discussed in Supplementary Note 6. As shown in Supplementary Figs. 2 and 3, modeling expected value at each timepoint allows us to quantify how fluctuations in game state impact likelihood of winning. Once again, the GP model outperforms a regularized logistic regression for each participant in our cohort (see Supplementary Fig. 13).

While our empirical expected value model successfully predicts each player's instantaneous prospects, it suffers from a key drawback: because it is conditioned on both players' observed (and coupled) strategies, it does little to disentangle the effects of each player's decisions on the trial's outcome. However, our task bears a strong resemblance to the class of differential games known as duels[14,20,22], in which players continuously evaluate options but choose only a single action. Along similar lines, we chose to analyze the expected value of the participant changing direction a final time and continuing on a straight-line trajectory thereafter. In keeping with the hockey analogy, this is equivalent to the instantaneous value of shooting the puck: the puck changes direction and moves under inertia thereafter. We estimated this value using the final change point of each trial as training data, with each opponent (two human players and the computer algorithm) modeled by a separate GP. This is justified by our assumption that, once participants have made their final move, all participant strategies are identical and trivial: the participant has no choices remaining. In reality, opponents in our game have imperfect information about whether participants have in fact committed to a final move, and so opponent beliefs about individual participants may be relevant in principle. Nonetheless, given the speed of the game and the following results, such an assumption serves as a useful baseline against which more elaborate assumptions about opponents' beliefs might be tested.

More formally, we estimated $Q_{\pi_{\text{final move}}}(\mathbf{s}, \omega)$, the participant's expected value of making the final direction change in state $s$ against opponent $\omega$. Here again, state $\mathbf{s}$ includes the positions and vertical velocities of both players, as well as the normalized time since the subject's last change point. Note the primary distinction between our action value models: The empirical EV model estimates the value of action $a$ to the participant at a particular moment, assuming both players follows their usual strategies $\pi$ thereafter. The final move EV again also estimates the value to the participant of changing direction, but assumes no change points thereafter.

Unsurprisingly, we found that there was a strong correlation between each subject's average expected value at final move and win rate against each opponent (Computer trials: $R = 0.31$, $p = 0.004$; trials against human opponent #1: $R = 0.665$, $p < 0.0001$; trials against human opponent #2: $R = 0.65$, $p < 0.0001$), demonstrating that this value is a good proxy for both win rate and shooter skill. Furthermore, when we plotted participants' estimated EV as a function of time within trial, we found that higher-scoring participants were more likely to locate their final change points during periods of high final move EV. Conversely, worse subjects showed the opposite pattern: they timed their final moves during periods of relatively low expected value (Fig. 5). Indeed, $\mathbb{E}|t_{\text{move}} - t_{\text{move}}^{\text{opt}}|$, the expected deviation between participants' actual and optimal final move times across trials, was significantly correlated with win rate against each opponent (computer trials: $R = -0.30$, $p = 0.0058$; human opponent #1 trials: $R = -0.48$, $p = 0.0004$; human opponent #2 trials: $R = -0.48$, $p = 0.0054$). In other words, the most successful participants were those who better concentrated their final change points within an optimal temporal window against each opponent.

However, it is also possible that higher-scoring participants might also create better shot opportunities for themselves in the early and middle stages of the trial, improving their overall prospects. For these participants, precise timing of the final move might conceivably be less important, due to their advantageous positioning early on in the trial. A schematic of expected value as a function of time in trial according to these two hypotheses (advantageous positioning vs. advantageous timing of final moves) are shown in Fig. 6a, b. If high-scoring participants are skilled at making decisions early on in the trial such that they place themselves in overall high expected value states, then one would observe vertical shifts in expected value between subjects near the end of each trial; conversely, a difference in the timing of final moves throughout trials need not be observed under this advantageous positioning hypothesis (Fig. 6a). By contrast, under an advantageous timing hypothesis, if timing one's final move is the most important factor in winning, we expect that the distribution of final move locations in time to discriminate between better and worse players (Fig. 6b). Observed data from the best and worst players against both a human opponent (Fig. 6c) and the computer opponent (Fig. 6d) show that timing one's final moves is, in fact, the decisive factor. While both the highest and lowest-scoring subjects experience similar expected values during the trial, higher-scoring subjects distribute their final move change points more effectively. This fits with the intuition of Fig. 5 and holds across our population (Supplementary Fig. 17).

## Discussion

Increasing interest in dynamic social interactions has necessitated a commensurate increase in the complexity of behavioral studies, but the methods used to analyze these new paradigms often lack the flexibility to handle the data produced. Here, we have shown that Gaussian Processes make it possible both to fit complex behavioral strategies and to forge links with the literature on reinforcement learning and game theory. Thus, our work is related to ideas in both inverse reinforcement learning[47–49], which seeks to estimate, rather than learn, policies and value functions capable of generating observed behavior and also the recent surge of interest in multi-agent reinforcement learning systems[50–52]. Our problem can be viewed as a limit of the game theory context in which decisions take place simultaneously in continuous time as well as dynamically against another opponent[5,52,53]. Our work stands to complement those results by

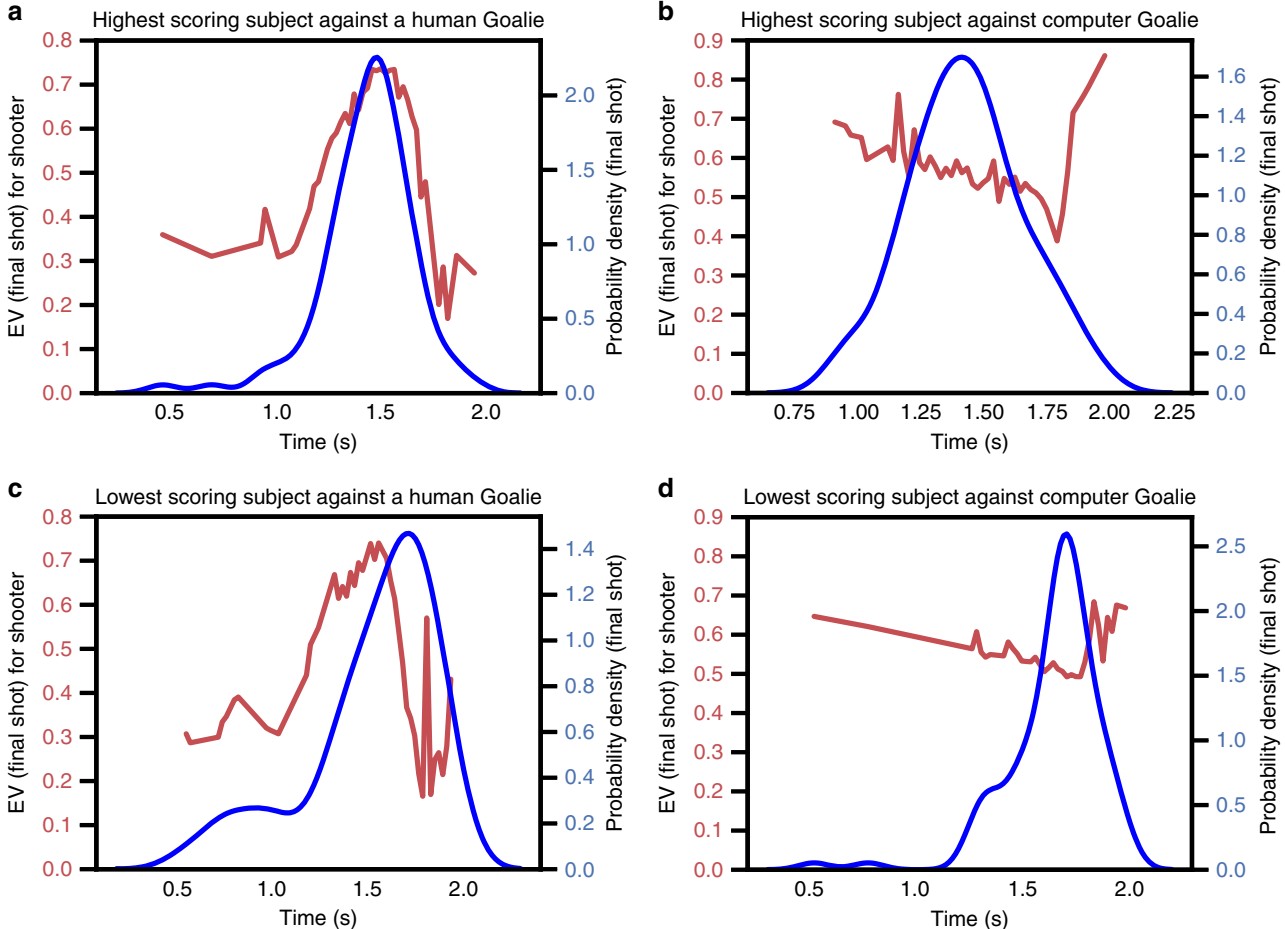

**Fig. 5** Skilled subjects time final moves in high expected value periods. Distribution of final change points (blue) in time compared with estimated expected value as a function of time (red) for four participants: **a** The participant with the highest score against a human opponent. **b** The participant with the highest score against the computer opponent. **c** The participant with the lowest score against a human opponent. **d** The participant with the lowest score against the computer opponent. More successful participants better aligned their final change points with critical periods in which the opponent was at a disadvantage

focusing on the dynamics that describe players' various decisions made throughout the task (in the policy model) as well as how valuable these actions are (in both action value models). This emphasis on the dynamic coupling of agents also bring us closer to real-world social interactions, in which decisions are based on coevolving exchanges.

There are several strengths to recommend our computational modeling framework. First, Bayesian estimation of continuous policy and value functions results in principled measures of uncertainty[32]. The resulting statistical inferences are better indicators of model fit than point estimates obtained from maximum likelihood methods. Second, differentiability of policies and value functions allows us to derive sensitivity estimates that quantify the coupling between agents, which we have shown can characterize individual differences in play on a variety of timescales. Third, modeling the joint distribution of both players allows us to perform counterfactual analyses (as shown in Supplementary Note 5) that dissociate the effects of player identity from those of game context—an intractable problem for most competing approaches[5]. Finally, dissociating policy and action value functions allows us to consider observed behavior without either assuming optimality or being able to calculate what optimal behavior should be[25,54].

Importantly, our approach of using flexible models like Gaussian Processes that capture the richness observed in real data

is not limited to a specific task. It generalizes readily to more than two agents, both cooperative and competitive contexts, and a wide variety of reward structures. All of these variants can be captured by simply enlarging the state space to accommodate the additional variables characterizing each agent, as is done in joint action learning algorithms[55,56]. Likewise, our data need not have been sampled densely or even at regular intervals, since Gaussian Processes have proven hugely influential in fields like ecology[39] and health data[44] where sparse observations are the norm. But our method is likely to prove most valuable for examinations of decision making in natural settings like shopping, foraging, or web browsing, where the number of covariates is large and the number of events (purchases, food items, clicks) is comparatively small. Nevertheless, the specific Gaussian Process formulation we have chosen has several limits. First, we have assumed that policies can be described as functions of instantaneous game state, which precludes situations where current behavior may depend on recent history. Second, as the number of agents grows, the required number of state variables grows exponentially. In this case, the data necessarily become even sparser and inferences more uncertain. Finally, even approximate GP methods can struggle when scaling to very large datasets. As a result, when data are abundant, parametric models like neural networks, which can be trained efficiently on subsets of data, may be more appropriate.

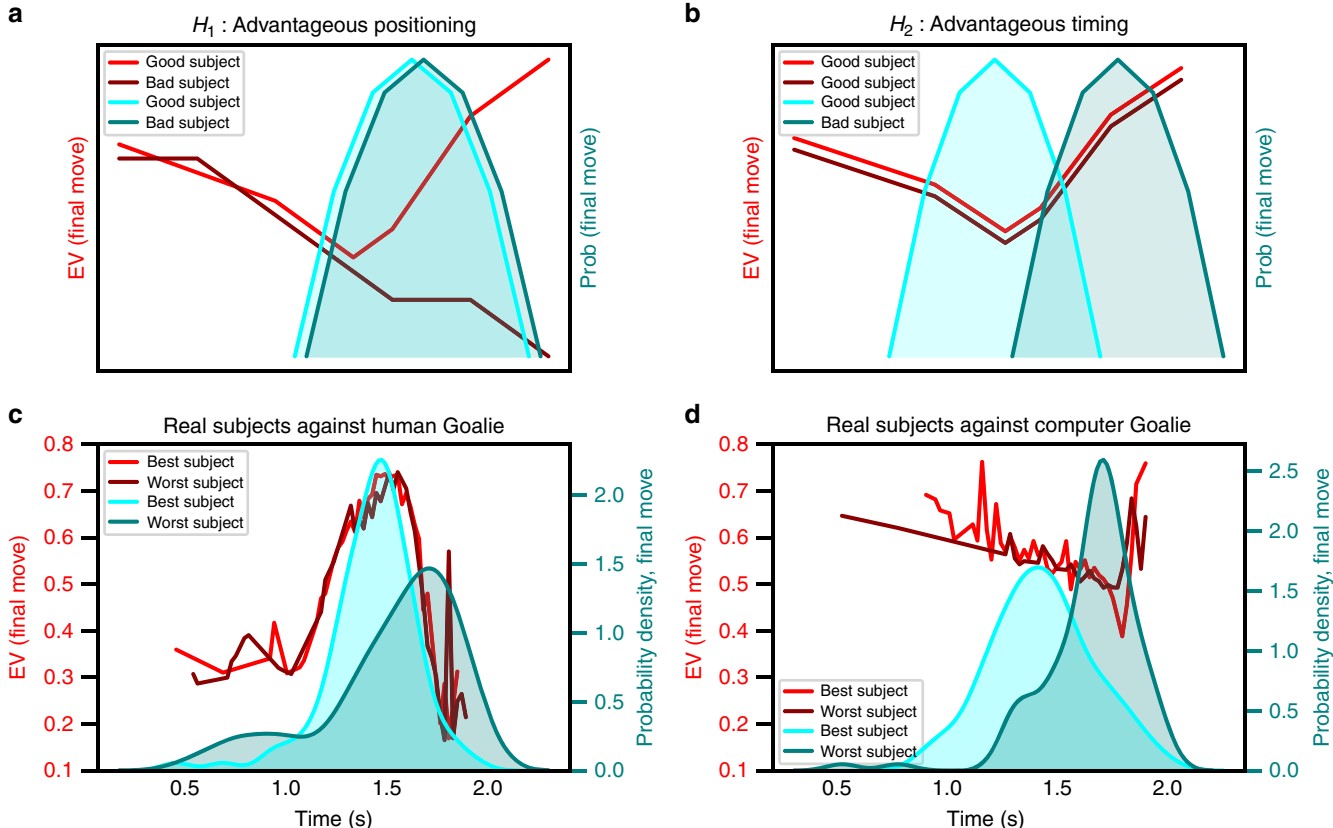

**Fig. 6** Data support advantageous timing over position hypothesis. Schematic of expected value and distribution of final change points as a function of win rate according to an advantageous position (H1) or timing hypothesis (H2). Red curves indicate the participant's expected value of the final move at each time. Blue distributions indicate the distributions of these final change points. **a** According to H1, higher-scoring participants are those who create better shot opportunites for themselves in the mid- and late-game, increasing their overall expected value. **b** According to H2, all participants experience roughly equal expected values during the trial, but higher-scoring participants are better at timing their final moves to exploit fleeting opportunities. **c, d** Observed results for the highest- and lowest-scoring participants against both of the human opponents (**c**) and the computer opponent (**d**). The results are consistent with H2 but not H1, implying that strategic timing is more important than positional advantage in task success

Our specific application yielded significant insights into humans' dynamic strategic interaction with both human and computer opponents. We found that participants exhibited a large variety of trajectories against both opponents, yet a majority of our participants demonstrated a heightened opponent sensitivity toward one particular goalie throughout the trial, and that subjects' sensitivity to opponent actions demonstrated substantial subject-level variability. The importance of these policy-derived metrics, particularly the sensitivities, is consistent with the findings of many groups that an ability to model the intentions of another agent plays a central role in human social interaction[8,57–59]. For our task, in which within-trial dynamics are more variable than across-trial changes in strategy, an analysis of variance showed that many sensitivities, including the baseline probability of switching and our sensitivity to opponent action metric, were relatively more trait-like than state-like, consistent with the idea that the underlying variability in our participant population is not in strategic heuristics but in the degree to which players' actions are coupled to one another. This decomposition of variance for continuous, task-related predictors can be used in future studies for systematically determining whether a given covariate characterizes a trait-like or state-like process, which is particularly important when investigating individual differences in the social sciences.

Finally, we showed that an analysis of participants' evolving prospects of winning easily distinguished between the track-then-guess heuristic of the computer opponent and the more complex human opponents. Such an analysis allows us not only to assess the degree to which a given moment in the trial is critical to a player's future prospects, but also how successful players are in seizing these opportunities. This is a result of the fact that action values are functions of both players' strategies. More specifically, by characterizing our task as a stopping problem, in which participants' objective was to time a single decision, we showed that an ability to accurately time one's final change point during periods of high expected value differentiated high-scoring from low-scoring participants. That effective players capitalize on momentary advantage is intuitive and relates to empirical work showing that expected value plays a role in strategic decision-making[3,10,60], as well as work which frames many common decisions in terms of foraging or search problems[61–63].

Perhaps most important for studies of social and decision neuroscience, our models suggest a natural set of variables of interest at a hierarchy of temporal scales. While the policies and action values we derive offer instantaneous regressors at the tens of milliseconds resolution of electrophysiology, including EEG and ECoG, these metrics can also be averaged at the trial and participant level for use with fMRI and PET. Providing computational frameworks for capturing complex temporal dynamics is crucial in learning and decision making[25,64,65]. The key advantage of our approach lies in an ability to identify both behavioral tipping points (high sensitivity of policy) and reward tipping points (large differences in action value) and distinguish between the two. This is particularly crucial in the analysis of neural data,

where one wishes to designate different types of cognitive events in addition to observational events (i.e., shifts in probability of winning without changes in action, or changes of mind)[66,67]. Taken together, our results and overall approach offer a new path to the use of more naturalistic paradigms in the study and modeling of social interaction.

## Methods

**Participants.** This study was approved by the Institutional Review Board of Duke University Medical Center. Data from 82 healthy volunteers (age range: 18–48 years; 45 females; 37 males) were included in the behavioral analyses. All participants gave written informed consent to participate in this experiment and were informed that no deception would be used throughout the experiment. Two long-term participants played the role of the human opponent in the penalty shot task, but each participant played against only one human opponent. The human opponents were not members of the study team and had no stake in the outcome of the study apart from maximizing their own compensation. Participants were told that on each trial they would play against either the human opponent they had met with during the consenting process, or against a computer algorithm. We emphasized to participants that deception would not be used in our task regarding who they were in fact playing against (i.e., when the task indicated they would play against the computer opponent next, they would indeed play against the computer). Our task was incentive-compatible: both the experimental participant and the human opponent were rewarded in monetary bonuses that were dependent on how frequently each player won.

Participants began the experiment with a 4 min practice block followed by three experimental blocks, each ~12 min long. Participants played as many trials as they could within each 12 min block, resulting in roughly 200 trials in total for each participant (~100 trials per opponent condition). At the beginning of each trial, each participant was prompted to center the joystick in order for the next trial to begin. A centered fixation cross was then presented for a jittered amount of time, ranging from 1.0 to 7.5 s. Following the fixation cross, the identity of the opponent on the upcoming trial (either "Computer" or the name of the human opponent) was displayed in centered text for 2 s. Each trial lasted roughly 1.5 s. Following the end of a trial, centered text displaying "WIN" or "LOSS" would appear on-screen, indicating the previous trial's outcome. Following the experiment, participants completed a post-task survey, were debriefed, and compensated.

**Puck and bar dynamics.** The puck was represented as a colored circle (of diameter $\frac{1}{64}$ of the screen width) and started each trial at normalized coordinate position $(-0.75, 0)$. The goal line was positioned at $x = 0.77$. The puck moved with constant horizontal velocity $v_p$ and vertical velocity $v_p u_t$, where $u_t \in [-1, 1]$ was the vertical joystick input at time $t$. The participant controlled only the vertical velocity of the puck. The puck was constrained to remain onscreen. At each time $t$, the coordinates of the puck were updated according to:

$$x_{t+1} = x_t + v_p \qquad (8)$$

$$y_{t+1} = y_t + v_p u_t. \qquad (9)$$

Both the human and computer opponents were identically represented on-screen by a vertical bar. The bar began each trial at $(0.75, 0)$, immediately to the left of the goal line, and could only move up or down. Unlike the puck, the opponent was able to accelerate: If the opponent maintained direction at near-maximal input ($|u| \in [0.8, 1]$) for three consecutive time steps, the bar's maximal velocity began to increase on the third step. That is, at each time step

$$v_\omega \leftarrow \frac{2}{3}\theta v_p \qquad (10)$$

$$\theta \leftarrow \begin{cases} \theta + 0.85, & \text{if accelerating} \\ 1, & \text{otherwise} \end{cases} \qquad (11)$$

$$y_{t+1} = y_t + v_\omega u_t. \qquad (12)$$

**Gaussian process model fitting.** Traditionally, performing full Bayesian inference in Gaussian processes has been prohibitive, with computation scaling as $\mathcal{O}(N^3)$, with $N$ the number of training data points. However, recent advances in approximate inference methods based on sparse collections of $M \ll N$ inducing points have reduced this cost to $\mathcal{O}(NM^2)$, making computation feasible for large datasets[33,34,45]. Here, we used GPFlow, a Gaussian process package based on the TensorFlow machine learning library, to fit separate Gaussian process classification models to data from each experimental participant[68]. Models were fit using the Sparse Variational Gaussian Process algorithm coded in GPFlow, using input variables as described in the text. We used 500 inducing points and trained for 200,000 iterations using the Adam optimizer[45,68,69] for both the policy and action value models. Altering these parameters did not materially change either the fitted GPs or their sensitivities (see Supplementary Figs. 15 and 16). Model hyperparameters were learned during the training run, an empirical Bayes approach[38]. We

used a train/test split of 80/20% at the timepoint level to evaluate each model's performance; test data were not used to select model parameters.

**Sensitivity metrics.** To capture the effect of small changes of input variables on our latent Gaussian Process $f$, we defined a sensitivity for each input variable as the (squared) norm of the GP gradient along that direction:

$$\nu_i(\mathbf{x}) = \left\| \sigma_i^{-1}(\mathbf{x}) \nabla_i f(\mathbf{x}) \right\|^2 \qquad (13)$$

with $i = 1 \dots 8$ indexing each predictor variable in $(\mathbf{s}, \omega)$ and $\sigma_i$ the local uncertainty in $\nabla_i f$. This can be motivated by noting that since $f$ is a GP, $\nabla f$ is as well (see Supplementary Note 2). Dividing a collection of squared Gaussian variables (one per observation) by their standard deviations results in a set of $\chi^2$ variables. Viewed another way, by normalizing by the uncertainty $\sigma_i$, we are downweighting highly uncertain gradients in our sensitivity measure (see Supplementary Note 4).

When we consider a total sensitivity to opponent actions, we combine sensitivities to opponent action and velocity into a single metric:

$$\varsigma = \left\| L^{-1} \nabla_{\tilde{x}} f(\mathbf{x}) \right\|^2 \qquad (14)$$

where $\varsigma$ is the opponent sensitivity metric, $\tilde{x} \equiv (y_{\text{opponent}}, v_{\text{opponent}})$ and $L$ is the Cholesky factor of the covariance of $\nabla_{\tilde{x}} f$ ($LL^T = \Sigma_x$). This is equivalent to combining the gradients for opponent position and velocity by first performing a PCA on these two coordinates and weighting each principal component equally in the calculation. As with the $\nu_i$ above, it can be shown that this index has a known distribution (noncentral $\chi^2$), allowing us to calculate uncertainty in the action sensitivity metric at each timepoint (see Supplementary Note 4).

**Reporting summary.** Further information on research design is available in the Nature Research Reporting Summary linked to this article.

## Data availability

The dataset generated and analysed during the current study are available on Open Science Framework (https://doi.org/10.17605/OSF.IO/EVFG5) (ref. [70]).

## Code availability

The analysis code that support the findings of this study have been made available at https://github.com/krm58/PenaltyShot_Behavior.

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

## Acknowledgements
We are thankful to the staff at Duke's Brain Imaging Analysis Center (BIAC) for assistance with data collection as well as NIH support (S10 OD 021480) for BIAC's super computing cluster. The GPU used for this research was donated by the NVIDIA Corporation. We are incredibly grateful to Shariq Iqbal for helpful discussions and key analysis insights. We also thank Dianna Amasino and Sam Dore for early data collection. Research reported in this publication was supported by a BD2K Career Development Award (K01-ES-025442) to J.M.P., an NIMH R01-108627 to S.H., and a National Science Foundation Graduate Research Fellowship under NSF GRFP DGE-1644868 and a Duke Dean's Graduate Fellowship to K.M.

## Author contributions
W.F.B. and S.H. designed research; K.M. performed data collection; K.M. and J.P. analyzed data; and K.M., S.H., and J.P. wrote the paper.

## Additional information

**Competing interests:** The authors declare no competing interests.

