## [Peer Review File · Nature Communications]

Reviewers' comments:

Reviewer #1 (Remarks to the Author):

This paper presents a sophisticated modeling framework for analyzing dynamic policy changes in a strategic decision making task. The model is descriptive: it captures policy changes without making any particular assumptions about the underlying computations. This is both a strength and a weakness. Its strength is that it offers an empirical window into decision behavior that is relatively unbiased by particular computational models. By the same token, its weakness is that it doesn't offer strong insights into the underlying decision process.

Major comments:

Comparison with logistic regression is measured in terms of log likelihood (Figures S9 and S10), but in Figure 2C performance is measured with AUC. It would be useful to have a consistent measure, or alternatively report both.

Was the logistic regression regularized? If so, were the hyperparameters optimized? These choices will strongly determine the performance of the model.

It's not clear what exactly we have learned about decision making from these analyses. The authors develop a number of clever analytical techniques (I particularly liked the sensitivity analysis), but it seemed like many of the conclusions boiled down to something like, "Here are some sources of variability across subjects." Ultimately one would like to ground this variability in some fundamental regularities, and ideally computational mechanisms. In my view, this is the main weakness of the paper. I realize that the authors do make some more specific claims about variability (e.g., that it's primarily driven by context rather than opponent) but I didn't feel that these insights were particularly earth-shaking, in part because they weren't contextualized in terms of "big" questions about decision making. In other words, what big questions do these insights help us answer?

Minor comments:

It would have been helpful to have page numbers.

Line 174: "They [GPs] are also equivalent to single-layer, fully-connected, infinitely wide neural networks" This equivalence is actually a special kind of GP. Although any GP can in principle be implemented with a neural network, not all GPs require full connectivity or infinite width.

Line 180: "More formally, a GP f is defined by..." f is the random function drawn from the GP, not the GP itself.

Eq. 1: technically, I think you want $f \sim \text{GP}(m,k)$, since the entire function is drawn from the GP.

Line 263: "the probability of switching, sensitivity to opponent action varied throughout the trial."
Missing comma.

Reviewer #2 (Remarks to the Author):

The authors propose a method to predict human behavior in strategic interaction settings with continuous feedback of opponent actions, and dynamic, co-evolving decisions by interacting agents. They study the trajectories that "penalty-takers" use to control their pucks in a "penalty shot task"

game. They estimate a policy function that returns the probability of a penalty-taker changing the vertical direction of the puck at each of a number of time points along the trajectory of the puck by using a Gaussian Process. They also evaluate a value-function for each action (changing the direction vs maintaining current path) for each individual at every time point along the puck trajectory to estimate how the expected value of winning (scoring a goal) is changing as the puck moves along its path. Their parameter-free model of reinforcement learning helps capture the strategies that penalty-takers use when playing such a game. While the authors do an impressive job analyzing their findings, I have some concerns regarding the scope of their contribution, which I have detailed in later sections.

General

The authors have a sensible idea for this project. Many existing models that attempt to predict human behavior in strategic interactions (from game theory) restrict their attention to games where players have limited action sets with discrete actions, and clearly defined turn taking between players. (Note however, that game theory has been carefully studied for about 80 years; even in early years people made progress in generalizing results to continuous strategies, like bidding in clock auctions, and in games over time and space such as duels.) The authors are attempting to model behavior in dynamic games where players take actions simultaneously from a large (possibly infinite) action space, while receiving continuous real-time feedback about the other player's actions. They claim such games are a better structure to analyze many natural social interactions. That's obviously true.

The specific game they study is the "penalty shot task", where one player (the "penalty taker") has a puck moving horizontally from the left side to the right side of a screen, attempting to score past the other player (the "goalie") into a goal on the right side of the screen. The penalty taker controls only the vertical direction and velocity of the puck using a joystick, while the goalie controls the vertical direction and velocity of a bar in front of the goal to try to block the puck using a joystick. 82 subjects played the role of penalty takers for approximately 200 trials of the game, with approximately 100 trials against 2 human opponent goalies, and the other approximately 100 trials against a computer. Each trial lasted about 1.5 seconds, equivalent to 94 - 96 discrete time points. The authors concentrate on finding a model that can capture the behavior of the penalty takers, trying to approximate the trajectory (in 2-dimensional space) taken by the puck as it traverses towards the goal. They simplify this challenge by identifying a policy function that prescribes, at each discrete time point, whether the penalty taker changes the vertical direction in which the puck is travelling, or allows it to continue in its current path.

At each time point, the policy function $n(a_t, s_t, \omega)$ takes in a vector of state variables (s) and the identity of the opponent - human or computer (ω), and returns a probability of taking an action a (stay the course, or change direction). If this probability is above some threshold (the authors use each participant's observed base rate of direction change as the threshold), then the policy function recommends changing direction. Otherwise, the policy function recommends continuing on the current trajectory. The vector of state variables at each time point contains: the x and y positions of the puck, the y position of the goalie's bar, the vertical velocities of the puck and the bar, the time since the occurrence of the last direction change, and the an opponent experience variable to reflect potential strategic adaptation.

The authors estimate a parameter-free model by linking this policy function to an underlying Gaussian Process (GP), a framework borrowed from the reinforcement learning literature. A GP is a distribution over functions, thus a sample from a GP is an entire function. The joint distribution of a dataset generated by such a function is d -dimensional multivariate normal. Thus, a dataset generated by a function $f(s, \omega)$ sampled from a GP is multivariate normal, and an inverse probit function applied to it returns a probability value, similar to the policy function $n(a_t, s_t, \omega)$. The authors model the dataset

of puck trajectories from the approximately 200 trials by each participant as being generated by such a function $f(s, \omega)$, and recover predicted actions (change of direction vs staying the course) at each time point. They cross-validate their model by recovering a parameter-free function $f(s, \omega)$ for each penalty taker by using 80% of their trials, and then use it to predict the trajectory for the remaining 20% of trials.

The following are the main findings in this paper:

1. The model is largely accurate in predicting time points along a puck trajectory in which a change point will occur. For the median participant, it has an AUC value of 94% of the change points from the 20% of trials held out for cross-validation purposes (Fig 2c). For every subject, these predictions are more accurate than a simple logistic regression model.

2. There are differences observed in the direction switch probabilities based on whether the participants were facing a human or a computer. Thus, penalty-takers' strategies are different based on whether the goalie was a human or a computer. The authors decompose this gap in switch probabilities into "opponent identity effects" - differences in how penalty takers perceive human vs computer opponents, and "opponent context effects" - differences in strategies used by human vs computer opponents that leads to differences in the underlying state variables. The "identity" effects come simply from the changes in ω , while the "context" effects change the underlying s , and both these effects directly and indirectly effect $f(s, \omega)$. Overall, they find that game state "context" effects are usually larger than pure "identity" effects. However, there is significant heterogeneity in the relative strength of these effects across the 82 penalty-takers. Furthermore, the average "identity" effect is heterogeneous across penalty-takers. Some penalty-takers have consistently higher switch probabilities against humans, while others exhibit consistently higher switch probabilities against computers.

3. Since the GP returns a continuous function $f(s, \omega)$ for each penalty-taker, the gradients of the GP with respect to the opponents' bar position and velocity returns the sensitivity of a penalty-taker's direction switching behavior to changes in the opponent's actions. These gradients are used to create a moment-by-moment sensitivity index. The gap in this sensitivity index against human vs computer opponents is decomposed into "identity" vs "context" effects. The authors find significant heterogeneity in the relative strength of these effects across the 82 penalty-takers, as well as significant heterogeneity in the overall gap across penalty-takers.

4. Similar to the sensitivity index for opponent actions, the authors create sensitivity indices for the direction switching probability to each of the state variables used to construct $f(s, \omega)$. The variance in each of these sensitivity indices (aggregated across trials and participant averages) is a combination of the variance within a trial (across time points), variance across trials within a particular penalty-taker, and variance across penalty-takers. The variance decomposition finds that the majority of the variance is within a trial across time points for most of the sensitivity indices. On the other hand, the probability of switching direction has more variance across penalty-takers than within a trial, so it is more like a personality trait than the sensitivity indices.

5. The authors use a new GP to generate an action-value function $Q(a|s, \omega)$ that predicts trial outcome using the state variables from earlier as well as the penalty-taker's observed action at each time point. Multiplying this action-value function with the policy function generated by the earlier GP gives the authors a value function returning the expected value of winning at every time point along the puck trajectory of each trial for every penalty-taker. Again, the GP process for the action-value function outperforms a logistic regression model for each penalty-taker.

There is a lot to like about this paper including the ambitious nature of the exercise, as well as the

attempt to study sensitivities of direction change probabilities to opponent actions and other state variables.

Main Concerns

The paper is clear in its objectives. There is a lot of jargon and notation that is special to computer science and multi-agent RL. The graphics are mostly effective at conveying what is going on however. Having said that, I have some concerns about the paper, most of which are from a big-picture "how much does this model really tell me?" perspective, but also some concerns about the benchmark models against which the authors compare their performance. In addition to these main concerns, I also enumerate some smaller concerns in the next section that would be nice for the authors to consider.

1. The authors compare the direction change predictions for each time point generated by their model estimated from the GP to predictions from a logistic regression model using the same state and opponent variables. Their model outperforms the predictions from the logistic regression model for each of the penalty takers in the sample. While this is encouraging, it is not particularly surprising. After all, the GP is searching across an entire class of functions to find one that best fits the generated dataset. On the other hand, the logistic regression model assumes a particular functional form for the underlying data generating process. For this particular type of dataset, it is not surprising that non-parametric methods are outperforming specific parametric models, and helps motivate how the authors are making a methodological contribution to the strategic interactions literature. However, they do not provide a motivation for why this particular non-parametric model is the best one. After all, instead of the GP model predicting direction changes of the puck at each time point, the authors could also consider a GP model that predicts the entire time series of the puck trajectory. Does their particular GP model also outperform other such potential non-parametric models? These are better benchmarks to calculate the relative performance of their model, not necessarily the logistic regression models they choose.

2. The only discussion of learning over trials, basically, is Fig S11 about changes in sensitivities from early to late. There should be other ways to illustrate how much policies are changing over time due to the RL type learning. Is the change nonlinear across trials (e.g. probably steep then flat as in most learning curves)?

Smaller Concerns

I also have some smaller suggestions for improvements that the authors could consider to refine this paper, or perhaps consider it for future research. However, not directly addressing these concerns should not be cause for rejecting the paper from publication.

1. The authors claim that their GP methodology to help identify human behavior in strategic interactions is not limited to a specific task, and is generalizable enough to accommodate a larger set of players, cooperative and competitive contexts, and different reward structures. They claim that behavior in these broader classes of games can be captured simply by changing the vector of state variables used to identify the policy function from the GP. I am a bit skeptical about this claim because the task here is so simple, and also the opponent behavior (except computer) is completely unmodelled. Could you unpack line 457 etc. and say a bit more about what domains would match closely to what you have done? Also please clarify when you say "our approach" do you mean GP in general?

2. In this paper, the authors are mainly attempting to model the behavior of the penalty-takers, but not necessarily the behavior of the human goalies (they set a pre-determined "track-then-guess" heuristic for computer goalies). It would be interesting to test the model when also considering a large

number of human goalies, and using the GP models to simultaneously predict direction changes in puck (and bar respectively) for both players simultaneously. This should not be that difficult because you basically have a candidate benchmark model already— the computer algorithm.

3. Typos: Page 31: 46 females + 37 males should make 83 participants, not 82.

4. Figure 2 has both black and grey dots but the caption does not explain what they encode (probably they are for the human vs computer trials).

5. Fig S9 what is on the x-axis?

6. Fig S12 caption is cut off in .pdf

7. Line 378 "Interestingly, while the types of trajectories generated by Participant 3 in both the human and computer opponent conditions look remarkably similar, expected values for these collections of trials evolve quite differently." I do not agree that they expected value paths look "quite different". Can you mention here how you are going to show this statistically (e.g. an identity effect in EV paths?)

8. Figure 7 "Conventions are as in A, B. Here, the "track-then-guess" heuristic of the computer opponent is apparent in the abrupt transition from a 50% unimodal distribution to a polarized bimodal distribution at the time of the opponent's last move." What do you mean by conventions? (labeling?). Also the transition is also evident in the human opponent, it just appears to happen earlier.

9. Line 432: "Our work stands to complement those results by focusing on the out-of-equilibrium dynamics that lead up to players' final moves." How do you know it is out of equilibrium since you have not computed any equilibrium?

10. "This emphasis on the dynamic coupling of agents also works to bring us closer to real-world social interactions, in which decisions are based on coevolving exchanges" There are many results on this, in continuous-time game theory. One example is "duels" in which players must approach each other in time and space. These were first explored in the late 1940s (see https://projecteuclid.org/download/pdf_1/euclid.Inms/1215453577). I suspect you have still made a contribution by modeling the penalty-kick game in terms of GP (and of course collecting data) but there may be a lot of mathematical results you need to cite and discuss as they may be relevant to how you interpret what is going on. For example, since it is convenient to discretize the time steps anyway, you can completely solve the game by backward induction with two states (locations of puck and bar). Since the game will have a mixed equilibrium a likely prediction is that the expected value distribution in space will be equal everywhere that is feasible to reach. This gives one method of thinking about behavioral "nonequilibrium" play.

11. One thing I always wonder about experiments with algorithmic opponents is—what are subjects told, and what do they think, the computers are doing? The same issue arises with a human opponent too; in a sense, they are trying to figure out what the human experimenters programmed the computer to do. My hunch is that there is a lot more learning about the computer strategy by the subjects over time than about humans (either because the human opponents are more stationary, or because they may be very cognizant that the computer was programmed). I do not like to second-guess design choices but if you have any information about what the subjects thought the computers (and other humans too) were doing that would be good to know, or could be added as debriefing in future data collection.

We appreciate both reviewers' thoughtful feedback on our work. Our manuscript has been substantially revised to offer both stronger links to the literature and new insights into the underlying decision processes involved in our task. In particular, we include a significant new analysis (detailed below) suggested by comments from Reviewer 2: Inspired by the literature on duels, we modeled the timing of participants' final change points. As we show in a new section ("Expected Value of Making One's Final Move") and Figures 5 and 6, this approach shows clearly which factors contribute to higher-scoring subjects' success and offers an intuitive means of characterizing player strategies based on our method.

In addition, we have expanded our discussion to provide closer links to other relevant literature, as suggested by both reviewers. For clarity of exposition, we have also moved some more technical results to supplementary material.

We hope both reviewers agree that these revisions significantly enhance the strength and impact of the work. Responses to both reviewers are included inline (in blue) below, with references to line numbers in the revised manuscript included where applicable.

Reviewer #1:

This paper presents a sophisticated modeling framework for analyzing dynamic policy changes in a strategic decision making task. The model is descriptive: it captures policy changes without making any particular assumptions about the underlying computations. This is both a strength and a weakness. Its strength is that it offers an empirical window into decision behavior that is relatively unbiased by particular computational models. By the same token, its weakness is that it doesn't offer strong insights into the underlying decision process.

We appreciate the reviewer's positive assessment of our work. We also agree that our previous manuscript left some gap between our results and an intuitive understanding of the decision process in the task.

Thus, inspired by a suggestion of Reviewer 2, we performed a new analysis that we believe gives significant new insight along these lines. In this new analysis, we considered a restricted "duel" scenario in which participants (who controlled the puck) chose when to execute their last change point of the trial. Using the same Gaussian Process approach, we modeled the expected value of this action and found that:

- Higher-scoring subjects distributed their final change points around moments of high expected value, while subjects with lower scores either distributed change points too broadly or centered their distributions around lower-EV moments in time. This is not unexpected — better participants should have higher EV for a final move — but it offers a clear, intuitive account of this key individual difference in our Task (Figure 5).
- While participants' showed substantial variability in *when* they placed their final change points, they displayed negligible differences in what we might call "positional advantage." That is, we might hypothesize that participants could be successful *either* by accurately

timing their final moves *or* by making strategic moves during the trial that increased the value of their future options across the board. We found strong evidence for the former and against the latter. High-scoring subjects and low-scoring subjects clearly differed in their ability to strategically time final moves but showed very similar expected value curves as a function of time in trial. Thus we were able to disambiguate two possible accounts of what makes for successful play in the task.

These findings are detailed in a new results section titled "Expected Value of Making One's Final Move," along with new Figures 5 and 6. We hope the reviewer will agree that these results, which build on our previous analysis, are more intuitive and provide additional insights into players' decision strategies.

Comparison with logistic regression is measured in terms of log likelihood (Figures S9 and S10), but in Figure 2C performance is measured with AUC. It would be useful to have a consistent measure, or alternatively report both.

We thank the reviewer for this suggestion. Throughout the paper and in the supplement, we report results from logistic regressions and gaussian processes in terms of AUC (lines 219-221; 347-349 and Figures 2, S12, S13).

Was the logistic regression regularized? If so, were the hyperparameters optimized? These choices will strongly determine the performance of the model.

The logistic regression in the original manuscript was not regularized, since there were 8 predictor variables and over 1 million observations. In this version of the manuscript, we implemented a logistic regression with an L1 penalty (lines 219-221; 347-349), though as expected given the number of data points, the regularization did not change the outcome. In fact, 52 of our 82 subjects had all coefficients present in their best-fitting model.

It's not clear what exactly we have learned about decision making from these analyses. The authors develop a number of clever analytical techniques (I particularly liked the sensitivity analysis), but it seemed like many of the conclusions boiled down to something like, "Here are some sources of variability across subjects." Ultimately one would like to ground this variability in some fundamental regularities, and ideally computational mechanisms. In my view, this is the main weakness of the paper. I realize that the authors do make some more specific claims about variability (e.g., that it's primarily driven by context rather than opponent) but I didn't feel that these insights were particularly earth-shaking, in part because they weren't contextualized in terms of "big" questions about decision making. In other words, what big questions do these insights help us answer?

We are sympathetic to the reviewer's concern here. We have attempted to address this in two ways: First, we have expanded our introduction and discussion to better place our task in the context of continuous games, as suggested by Reviewer 2. Second, as explained above, we

have added a substantial new analysis that uses the Gaussian Process approach of our previous manuscript to examine how participants allocate their final moves. As noted in the results and discussion (lines 321-420; 500-519), this produces a quantitative dissociation between positional and timing advantages that we expect to prove useful for the analysis of similar games, especially those with realistic reaction time constraints. Moreover, we believe that the combination of our task and analysis offer a fruitful platform for future studies of the neuroscientific basis of dynamic decisions.

It would have been helpful to have page numbers.

We agree. These have been added.

Line 174: "They [GPs] are also equivalent to single-layer, fully-connected, infinitely wide neural networks" This equivalence is actually a special kind of GP. Although any GP can in principle be implemented with a neural network, not all GPs require full connectivity or infinite width.

The reviewer is, of course, correct. Our revised manuscript (lines 184-187) now reads:

And while some types of GPs are equivalent to infinitely-wide, single-layer neural networks, they have been shown to outperform neural networks in avoiding overfitting on small to moderate datasets.

Line 180: "More formally, a GP f is defined by..." f is the random function drawn from the GP, not the GP itself.

We thank the reviewer for this correction: we have edited this point in the manuscript (Line 194) to specify that $f(x)$ is a random function drawn from the GP.

Eq. 1: technically, I think you want $f \sim \text{GP}(m,k)$, since the entire function is drawn from the GP.

We appreciate the correction. We have noted this in the text.

Line 263: "the probability of switching, sensitivity to opponent action varied throughout the trial." Missing comma.

Corrected.

Reviewer #2 (Remarks to the Author):

The authors propose a method to predict human behavior in strategic interaction settings with continuous feedback of opponent actions, and dynamic, co-evolving decisions by interacting agents. They study the trajectories that "penalty-takers" use to control their pucks in a "penalty shot task" game. They estimate a policy function that returns the probability of a penalty-taker

changing the vertical direction of the puck at each of a number of time points along the trajectory of the puck by using a Gaussian Process. They also evaluate a value-function for each action (changing the direction vs maintaining current path) for each individual at every time point along the puck trajectory to estimate how the expected value of winning (scoring a goal) is changing as the puck moves along its path. Their parameter-free model of reinforcement learning helps capture the strategies that penalty-takers use when playing such a game. While the authors do an impressive job analyzing their findings, I have some concerns regarding the scope of their contribution, which I have detailed in later sections.

General

The authors have a sensible idea for this project. Many existing models that attempt to predict human behavior in strategic interactions (from game theory) restrict their attention to games where players have limited action sets with discrete actions, and clearly defined turn taking between players. (Note however, that game theory has been carefully studied for about 80 years; even in early years people made progress in generalizing results to continuous strategies, like bidding in clock auctions, and in games over time and space such as duels.) The authors are attempting to model behavior in dynamic games where players take actions simultaneously from a large (possibly infinite) action space, while receiving continuous real-time feedback about the other player's actions. They claim such games are a better structure to analyze many natural social interactions. That's obviously true.

The specific game they study is the "penalty shot task", where one player (the "penalty taker") has a puck moving horizontally from the left side to the right side of a screen, attempting to score past the other player (the "goalie") into a goal on the right side of the screen. The penalty taker controls only the vertical direction and velocity of the puck using a joystick, while the goalie controls the vertical direction and velocity of a bar in front of the goal to try to block the puck using a joystick. 82 subjects played the role of penalty takers for approximately 200 trials of the game, with approximately 100 trials against 2 human opponent goalies, and the other approximately 100 trials against a computer. Each trial lasted about 1.5 seconds, equivalent to 94 - 96 discrete time points. The authors concentrate on finding a model that can capture the behavior of the penalty takers, trying to approximate the trajectory (in 2-dimensional space) taken by the puck as it traverses towards the goal. They simplify this challenge by identifying a policy function that prescribes, at each discrete time point, whether the penalty taker changes the vertical direction in which the puck is travelling, or allows it to continue in its current path.

At each time point, the policy function π (at t , s_t , ω) takes in a vector of state variables (s) and the identity of the opponent - human or computer (ω), and returns a probability of taking an action a (stay the course, or change direction). If this probability is above some threshold (the authors use each participant's observed base rate of direction change as the threshold), then the policy function recommends changing direction. Otherwise, the policy function recommends continuing on the current trajectory. The vector of state variables at each time point contains:

the x and y positions of the puck, the y position of the goalie's bar, the vertical velocities of the puck and the bar, the time since the occurrence of the last direction change, and the an opponent experience variable to reflect potential strategic adaptation.

The authors estimate a parameter-free model by linking this policy function to an underlying Gaussian Process (GP), a framework borrowed from the reinforcement learning literature. A GP is a distribution over functions, thus a sample from a GP is an entire function. The joint distribution of a dataset generated by such a function is d-dimensional multivariate normal. Thus, a dataset generated by a function $f(s, \omega)$ sampled from a GP is multivariate normal, and an inverse probit function applied to it returns a probability value, similar to the policy function $\pi(a_t, s_t, \omega)$. The authors model the dataset of puck trajectories from the approximately 200 trials by each participant as being generated by such a function $f(s, \omega)$, and recover predicted actions (change of direction vs staying the course) at each time point. They cross-validate their model by recovering a parameter-free function $f(s, \omega)$ for each penalty taker by using 80% of their trials, and then use it to predict the trajectory for the remaining 20% of trials.

The following are the main findings in this paper:

1. The model is largely accurate in predicting time points along a puck trajectory in which a change point will occur. For the median participant, it has an AUC value of 94% of the change points from the 20% of trials held out for cross-validation purposes (Fig 2c). For every subject, these predictions are more accurate than a simple logistic regression model.
2. There are differences observed in the direction switch probabilities based on whether the participants were facing a human or a computer. Thus, penalty-takers' strategies are different based on whether the goalie was a human or a computer. The authors decompose this gap in switch probabilities into "opponent identity effects" - differences in how penalty takers perceive human vs computer opponents, and "opponent context effects" - differences in strategies used by human vs computer opponents that leads to differences in the underlying state variables. The "identity" effects come simply from the changes in ω , while the "context" effects change the underlying s , and both these effects directly and indirectly effect $f(s, \omega)$. Overall, they find that game state "context" effects are usually larger than pure "identity" effects. However, there is significant heterogeneity in the relative strength of these effects across the 82 penalty-takers. Furthermore, the average "identity" effect is heterogeneous across penalty-takers. Some penalty-takers have consistently higher switch probabilities against humans, while others exhibit consistently higher switch probabilities against computers.
3. Since the GP returns a continuous function $f(s, \omega)$ for each penalty-taker, the gradients of the GP with respect to the opponents' bar position and velocity returns the sensitivity of a penalty-taker's direction switching behavior to changes in the opponent's actions. These gradients are used to create a moment-by-moment sensitivity index. The gap in this sensitivity index against human vs computer opponents is decomposed into "identity" vs "context" effects. The authors find significant heterogeneity in the relative strength of these effects across the 82 penalty-takers, as well as significant heterogeneity in the overall gap across penalty-takers.

4. Similar to the sensitivity index for opponent actions, the authors create sensitivity indices for the direction switching probability to each of the state variables used to construct $f(s, \omega)$. The variance in each of these sensitivity indices (aggregated across trials and participant averages) is a combination of the variance within a trial (across time points), variance across trials within a particular penalty-taker, and variance across penalty-takers. The variance decomposition finds that the majority of the variance is within a trial across time points for most of the sensitivity indices. On the other hand, the probability of switching direction has more variance across penalty-takers than within a trial, so it is more like a personality trait than the sensitivity indices.

5. The authors use a new GP to generate an action-value function $Q(a|s, \omega)$ that predicts trial outcome using the state variables from earlier as well as the penalty-taker's observed action at each time point. Multiplying this action-value function with the policy function generated by the earlier GP gives the authors a value function returning the expected value of winning at every time point along the puck trajectory of each trial for every penalty-taker. Again, the GP process for the action-value function outperforms a logistic regression model for each penalty-taker. There is a lot to like about this paper including the ambitious nature of the exercise, as well as the attempt to study sensitivities of direction change probabilities to opponent actions and other state variables.

Main Concerns

The paper is clear in its objectives. There is a lot of jargon and notation that is special to computer science and multi-agent RL. The graphics are mostly effective at conveying what is going on however. Having said that, I have some concerns about the paper, most of which are from a big-picture "how much does this model really tell me?" perspective, but also some concerns about the benchmark models against which the authors compare their performance. In addition to these main concerns, I also enumerate some smaller concerns in the next section that would be nice for the authors to consider.

We thank Reviewer 2 for the thorough and careful assessment of our work. We agree with both Reviewers 1 and 2 that the links to broader questions about decision-making could be strengthened. Thus, as detailed in our reply to Reviewer 1, and as suggested by the literature on continuous games suggested here by Reviewer 2, we have added a significant new analysis (detailed in the section "Expected Value of Making One's Final Move" and in Figures 5 and 6) that leverages our modeling approach to give an intuitive account of strategic differences between more and less successful participants. This analysis thus speaks to broader considerations of local (timing) versus global (positional) advantage in extended decision-making. We hope that the reviewer will agree that these new results substantially strengthen the appeal of the work.

1. The authors compare the direction change predictions for each time point generated by their model estimated from the GP to predictions from a logistic regression model using the same

state and opponent variables. Their model outperforms the predictions from the logistic regression model for each of the penalty takers in the sample. While this is encouraging, it is not particularly surprising. After all, the GP is searching across an entire class of functions to find one that best fits the generated dataset. On the other hand, the logistic regression model assumes a particular functional form for the underlying data generating process. For this particular type of dataset, it is not surprising that non-parametric methods are outperforming specific parametric models, and helps motivate how the authors are making a methodological contribution to the strategic interactions literature. However, they do not provide a motivation for why this particular non-parametric model is the best one. After all, instead of the GP model predicting direction changes of the puck at each time point, the authors could also consider a GP model that predicts the entire time series of the puck trajectory. Does their particular GP model also outperform other such potential non-parametric models? These are better benchmarks to calculate the relative performance of their model, not necessarily the logistic regression models they choose.

We agree with the reviewer that the result of our GP/logistic regression comparison is neither impressive nor surprising, given the complexity of each model class. However, we do wish to point out that just such a logistic regression approach still has a great deal of currency in the cognitive neuroscience literature, so the comparison remains useful for those more familiar with that technique.

The reviewer also offers a very interesting suggestion, which we also considered: modeling entire trajectories using the GP approach. In fact, we have pursued an approach of modeling entire trajectories in a related paradigm elsewhere (Iqbal and Pearson arXiv:1702.07319; Iqbal, Yin, Drucker, Kuang, Gariepy, Platt, and Pearson (under review)). As for the particular suggestion of using Gaussian processes for this purpose, there are two technical hurdles to consider:

1. Since we wish to model both players' trajectories jointly, this would be a multi-output GP. And while there are methods for doing this, it does increase both the computational complexity of the model and the cognitive complexity for readers.
2. Because draws from a GP will concentrate around their mean functions, we would expect actual trajectories to exhibit this property. However, as our data show, trajectories exhibit a wide variety of shapes, which would require either inferring a single mean function around 0 with high posterior uncertainty (resulting in many trajectories that look entirely unrealistic) or using a mixture of GPs, one for each trajectory "subtype." The latter is probably a productive approach, but we consider it outside the scope of this work.

More importantly, as we detail in lines 170-176:

Our selection of model was guided by three requirements: First, the model should be flexible enough to capture the rich diversity of player behavior. Second, the model should appropriately handle a small number of change points (~ 4.6%) with an input space of moderate dimension.

And third, the model should avoid overfitting while providing a principled estimate of uncertainty. For these reasons, we fit each participant's data using a Gaussian Process (GP) classification model.

That is, we think it less important whether the GP produces the best fits of any possible model class (though we would argue that Figure 2 provides some evidence we are doing quite well) than whether we have sufficiently good fits to facilitate accurate inference. And one of the key benefits of the GP is, as we note in the text, its ability to quantify uncertainty. Put another way, we do not believe the contribution of this work lays in proving the superiority of GPs as a model class, but in illustrating that GPs *can* do well fitting complex data and *more importantly* that they allow us to answer questions about uncertainty and sensitivity that are difficult to address using other approaches.

2. The only discussion of learning over trials, basically, is Fig S11 about changes in sensitivities from early to late. There should be other ways to illustrate how much policies are changing over time due to the RL type learning. Is the change nonlinear across trials (e.g. probably steep then flat as in most learning curves)?

This is another excellent point, and one that we considered. In our current model, we include as part of the game state an “opponent experience” variable that ranges from 0 (first trial against this opponent, human or computer) to 1 (last trial against this opponent). We would expect that, if participants’ strategies are changing over the course of the experiment, their policy functions should be sensitive to this variable. However, using an automatic relevance determination approach in which we also optimize the parameters of the GP kernels during learning, we found that the kernel length scales corresponding to opponent experience were almost always large (see Supplementary Figure 18), indicating a negligible effect of opponent experience on policy. In other words, given even brief experience to the game prior to playing inside the scanner, participants’ strategies quickly stabilized. This is not what we were expecting, but given that trajectories from participants’ early- and late-session trials show no obvious difference (see Supplementary Figure 19), it accords with the data.

We have also made note of this finding in our revised manuscript (lines 222-233):

Smaller Concerns

I also have some smaller suggestions for improvements that the authors could consider to refine this paper, or perhaps consider it for future research. However, not directly addressing these concerns should not be cause for rejecting the paper from publication.

1. The authors claim that their GP methodology to help identify human behavior in strategic interactions is not limited to a specific task, and is generalizable enough to accommodate a larger set of players, cooperative and competitive contexts, and different reward structures. They claim that behavior in these broader classes of games can be captured simply by changing the vector of state variables used to identify the policy function from the GP. I am a bit

skeptical about this claim because the task here is so simple, and also the opponent behavior (except computer) is completely unmodelled. Could you unpack line 457 etc. and say a bit more about what domains would match closely to what you have done? Also please clarify when you say “our approach” do you mean GP in general?

We appreciate the suggestion. More specifically, we expect that our approach, using GPs to model policy functions, will work when policies are well-described as simple functions of instantaneous state. This precludes, for instance, situations with long-lived correlations with past state or those in which the number of agents is sufficiently large that the resulting state space is high-dimensional and the data comparatively sparse. From another direction, even approximate GP inference methods are limited when data are massive, so data sets of millions of data points are likely feasible when policies are smooth, while those with billions (or very complex policies) will not be well-modeled.

The relevant section (lines 469-478) addressing this point now reads:

Nevertheless, the specific Gaussian Process formulation we have chosen has several limits. First, we have assumed that policies can be described as functions of instantaneous game state, which precludes situations where current behavior may depend on recent history. Second, as the number of agents grows, the required number of state variables grows exponentially. In this case, the data necessarily become even sparser and inferences more uncertain. Finally, even approximate GP methods can struggle when scaling to very large data sets. As a result, when data are abundant, parametric models like neural networks, which can be trained efficiently on subsets of data, may be more appropriate.

2. In this paper, the authors are mainly attempting to model the behavior of the penalty-takers, but not necessarily the behavior of the human goalies (they set a pre-determined “track-then-guess” heuristic for computer goalies). It would be interesting to test the model when also considering a large number of human goalies, and using the GP models to simultaneously predict direction changes in puck (and bar respectively) for both players simultaneously. This should not be that difficult because you basically have a candidate benchmark model already—the computer algorithm.

We agree that this would be an interesting extension of the work. We have collected some data in which players alternate roles, which we are preparing to analyze for a future publication.

3. Typos: Page 31: 46 females + 37 males should make 83 participants, not 82.

Thank you. This has been changed (Line 540).

4. Figure 2 has both black and grey dots but the caption does not explain what they encode (probably they are for the human vs computer trials).

Figure 2 only contains black dots (in A) and shaded regions (in B), but they contain a level of transparency in the plot that makes some appear grey. We edited the caption of this figure to point this out.

5. Fig S9 what is on the x-axis?

The original Fig S9 is a violin plot of the difference in log likelihood between logistic regression and GP models for each subject. This figure has been replaced with a histogram that uses AUC (Figures S12,13).

6. Fig S12 caption is cut off in .pdf

This has been corrected.

7. Line 378 “Interestingly, while the types of trajectories generated by Participant 3 in both the human and computer opponent conditions look remarkably similar, expected values for these collections of trials evolve quite differently.” I do not agree that they expected value paths look “quite different”. Can you mention here how you are going to show this statistically (e.g. an identity effect in EV paths?)

This point has been removed from the main text to accommodate our new Final Move GP analysis results.

8. Figure 7 “Conventions are as in A, B. Here, the “track-then-guess” heuristic of the computer opponent is apparent in the abrupt transition from a 50% unimodal distribution to a polarized bimodal distribution at the time of the opponent’s last move.” What do you mean by conventions? (labeling?). Also the transition is also evident in the human opponent, it just appears to happen earlier.

Yes, by “Conventions are as in A,B”, we meant that the same conventions used in subplots A and B are similarly used in C,D. This figure has been moved to the supplement. We clarified this sentence to reflect “Label conventions” in new Figure S3. And yes, due to the nature of our task, at some undefined point (which depends on the play dynamics of both players), expected value must reach either 0 or 1, since each trials results in either a “win” or “loss” for each player. While this is seen in both players as Reviewer 2 notes, this transition is more abrupt for the expected values against the computer opponent.

9. Line 432: “Our work stands to complement those results by focusing on the out-of-equilibrium dynamics that lead up to players’ final moves.” How do you know it is out of equilibrium since you have not computed any equilibrium?

This is definitely a valid point; we have removed this sentence from the discussion to accommodate the incorporation of new text addressing the reviews.

10. “This emphasis on the dynamic coupling of agents also works to bring us closer to real-world social interactions, in which decisions are based on coevolving exchanges” There are many results on this, in continuous-time game theory. One example is “duels” in which players must approach each other in time and space. These were first explored in the late 1940s (see https://projecteuclid.org/download/pdf_1/euclid.Inms/1215453577). I suspect you have still made a contribution by modeling the penalty-kick game in terms of GP (and of course collecting data) but there may be a lot of mathematical results you need to cite and discuss as they may be relevant to how you interpret what is going on. For example, since it is convenient to discretize the time steps anyway, you can completely solve the game by backward induction with two states (locations of puck and bar). Since the game will have a mixed equilibrium a likely prediction is that the expected value distribution in space will be equal everywhere that is feasible to reach. This gives one method of thinking about behavioral “nonequilibrium” play.

We thank Reviewer 2 for this thoughtful point. As mentioned above, our new analysis was inspired by reformulating the task of the participant as a continuous-time *stopping problem*. We have also incorporated citations to this literature in both our introduction and discussion.

11. One thing I always wonder about experiments with algorithmic opponents is—what are subjects told, and what do they think, the computers are doing? The same issue arises with a human opponent too; in a sense, they are trying to figure out what the human experimenters programmed the computer to do. My hunch is that there is a lot more learning about the computer strategy by the subjects over time than about humans (either because the human opponents are more stationary, or because they may be very cognizant that the computer was programmed). I do not like to second-guess design choices but if you have any information about what the subjects thought the computers (and other humans too) were doing that would be good to know, or could be added as debriefing in future data collection.

We agree that this is an important point for social cognitive neuroscience work involving other agents (either human or algorithmic). We have added clarifying text regarding what participants were told regarding both opponents during the consenting and instructions process starting on Line 549-554. We have also added results from self-report data collected at the end of the experiment asking participants which opponent they thought was the better opponent: exactly half (N=41) of the participants reported they thought the human opponent was better, and thus the other half reported they thought the computer opponent was better. We mention in the revised manuscript that this self-report data suggests that the computer algorithm did, in fact, play at a level comparable to both human opponents. Further, (see Line 549-554), participants were told that there was absolutely no deception used in our experiment, so when they were told that they would play against a particular opponent in the following trial, they were given reason to believe that this was accurate.

However, as noted above, the automatic relevance determination analysis did not reveal effects of opponent experience for most participants, though this inference is conditioned on our choice

of model and kernel. We are likewise interested in what priors and learning assumptions humans bring to dealing with artificial agents, as this is a difficult problem worthy of future work.

****REVIEWERS' COMMENTS:**

Reviewer #1 (Remarks to the Author):

The authors have adequately responded to my comments. The new analyses are informative about the underlying decision strategies, though I'm not sure they get us that much closer to better models of decision making outside this particular task. That being said, my opinion is somewhat subjective and I think the technical contributions of the paper in terms of descriptive modeling are sufficient to merit publication.

Reviewer #2 (Remarks to the Author):

(former Ref 2) This is a responsive revision which is much appreciated. I like the Figs 5-6 analyses and decomposition of position versus final move execution. The graphs are nice and clearly support one hypothesis (H1). This is a nice task and can see it becoming paradigmatic for a lot of social interaction type experiments.

We appreciate both reviewers' feedback on our work. Our revised manuscript has been edited to fit with Nature Communications' formatting guidelines. In particular, the formatting requests made in the approval letter pertaining to the following sections have been addressed: Title Page, Main Text, Language and Style, Methods and Data, End Notes, Display Items, Supplementary Information. In addition, responses to both reviewers are included inline (in blue) below.

Reviewer #1 (Remarks to the Author):

The authors have adequately responded to my comments. The new analyses are informative about the underlying decision strategies, though I'm not sure they get us that much closer to better models of decision making outside this particular task. That being said, my opinion is somewhat subjective and I think the technical contributions of the paper in terms of descriptive modeling are sufficient to merit publication.

We thank Reviewer 1 for suggesting the additional analyses, which have contributed to the development of our thinking and the current results.

Reviewer #2 (Remarks to the Author):

(former Ref 2) This is a responsive revision which is much appreciated. I like the Figs 5-6 analyses and decomposition of position versus final move execution. The graphs are nice and clearly support one hypothesis (H1). This is a nice task and can see it becoming paradigmatic for a lot of social interaction type experiments.

We sincerely appreciate Reviewer 2's comments and definitely agree that the newly added figures 5 and 6 and the corresponding analyses contributed to the manuscript overall. We indeed hope that other researchers find our paradigm helpful in future social cognition experiments.